

# Assessment of Future Precipitation Changes in Mediterranean Climate Regions from CMIP6 ensemble

Patricia Tarín-Carrasco[1], Desislava Petrova[1], Laura Chica-Castells[1], Jelena Lukovic[2], Xavier Rodó[1,3] and Ivana Cvijanovic[1]

[1]Barcelona Institute for Global Health (ISGlobal), Climate and Health Programme, Barcelona, Catalonia, Spain
[2]University of Belgrade, Faculty of Geography, Belgrade, Serbia
[3]ICREA, Barcelona, Catalonia, Spain

*Correspondence to*: Patricia Tarín-Carrasco (patricia.tarin@isglobal.org)

**Abstract.** Previous studies have indicated a large model disagreement in the future projections of precipitation changes over the regions featuring Mediterranean climate. Many of these highly populated regions have been experiencing major droughts in the recent decades, raising concerns about future precipitation changes and their impacts. Here we investigate precipitation projections across the five Mediterranean climate regions in the CMIP6 ensemble, and study their respective model agreements on the sign of future precipitation changes. We focus on the period 2050-2079 relative to 1970-1999, and consider two climate change scenarios (ssp2-4.5 and ssp5-8.5) over the Mediterranean Basin (MED), California (CAL), the central coast of Chile (SAA), the Cape Province area of South Africa (SAF) and southwest Australia (AUS).

The CMIP6 ensemble mean suggests that annual mean cumulative precipitation will decrease over all the regions studied with the exception of northern California. In most cases, this decline is primarily attributed to a reduction in winter precipitation, except over the Mediterranean Basin, where the most significant decrease occurs in autumn. The model agreement on the sign of future precipitation changes is generally high over the regions and seasons where the ensemble mean indicates the precipitation decline in the future, and low over the regions showing the precipitation increase or no change. Specifically, the model agreement is low in southern California during all seasons, in northern Mediterranean during winter and autumn, and in southwest Australia during austral summer and autumn. CMIP6 ensemble means also indicate that the consecutive dry days (CDD) will increase in the future in all regions, but again the model agreement on this increase is low over southern and central California, the southern Mediterranean, and parts of southwest Australia. Similarly, the ensemble mean consecutive wet days (CWD) indicates a decrease in all regions, with weak model agreement on the sign of future changes over CAL, northeast AUS and part of the MED region. The ensemble mean maximum one-day precipitation increases over all the regions, the most over the parts of southwest Australia and the Mediterranean.

We conclude that despite substantial improvements to the new CMIP6 generation of models, the intermodel differences in future projections of precipitation changes continue to be high across parts of California, the Mediterranean Basin and southwest Australia. Impact studies need to account for these uncertainties and consider the whole intermodel range of projected precipitation changes.





**1 Introduction**

Mediterranean climate (MedClim) regions - the Mediterranean Basin, California, the central coast of Chile, the Cape Province
area of South Africa and southwest Australia - are characterised by warm and dry summers, and mild and wet winters.
Hydroclimate seasonality is the strongest characteristic of these regions, as the majority of rainfall occurs during the cold
season, resulting in large precipitation variability throughout the year. MedClim regions are located on the west sides of
continents at latitudes between 30° and 40°, and their weather is seasonally influenced both by tropical and mid-latitude
atmospheric circulation patterns. Downward motion from the subsiding branches of the Hadley circulation favours stable and
dry weather over the MedClim regions most of the year except for the winter season when the mid-latitude storms are more
likely to make an appearance (Frierson et al., 2007). However, since the MedClim regions are also situated on the coast, the
adjacent seas and oceans can also act as sources of moisture for cyclone development during the warmer seasons (Lionello et
al., 2006).

The highly populated MedClim regions are especially vulnerable to precipitation decline. Spain, northern Italy, southern
California, Chile and Australia have all been showing decreasing trends in annual precipitation amounts (Deitch et al., 2017).
Droughts in the Mediterranean Basin have increased in frequency and intensity in the last decades (Kelley et al., 2015; Vicente-
Serrano et al., 2014) and since the late 2021, southern Europe has been suffering a severe drought that continued into the
summer of 2023 (Toreti et al., 2023c). In May 2023, at the end of the typical wet season, the total water reservoir storage in
Spain fell below 50% of its total capacity (Toreti et al., 2023c). The Tagus river, providing irrigation to the south-eastern
region of Spain, dropped to the lowest streamflow on record in 2023, resulting in the Spanish government limiting the water
transfers to the south-eastern Levante, and causing substantial agricultural and economic difficulties (Toreti et al., 2023c). In
addition, in 2022 the Spanish government failed to meet the transfer of the annual water flow agreed in the Convenio de
Albufeira, the bilateral water agreement with Portugal, leading to tensions on both sides about the future management of this
resource (Ministry for Ecological Transition and Demographic Challenge, 2022).  Globally, economic losses due to drought
have been increasing, with the economic loss from the last decade (2010-2019) being the largest out of the last four decades
(Douris et al., 2023).  In the EU, from 1990 to 2016, annual loss due to droughts were reported to be close to 9 billion €, with
the highest losses in Spain (1.5 billion € year⁻¹), Italy (1.4 billion € year⁻¹) and France (1.2 billion € year⁻¹), depending on the
region, between 39-60% of the losses relate to agriculture and 22-48% to the energy sector (Cammalleri et al., 2020).

Climate change, in combination with growing population, is expected to further increase the water scarcity in MedClim regions
(Cramer et al., 2018; Lau et al., 2013; Kharin et al., 2007). For the Mediterranean Basin, often referred to as a climate change
"hot spot" (Giorgi 2006), Lionello and Scarascia (2018) suggest that for each 1°C of global warming, mean rainfall will
decrease by about 4% in much of the region. Cramer et al., (2018) further highlights the increasing vulnerability and risk posed
by climate change in the Mediterranean Basin in the next decades, in particular in the coastal areas of the eastern and southern
Mediterranean, where the increasing population and urbanisation will not only lead to higher water demand, but also to further
deterioration of water quality.





Projections of future precipitation changes are still highly uncertain over many MedClim regions, thus impeding the assessment of the overall impacts on water supply and availability. Studies using the Climate Model Intercomparison Project Phase 5 (CMIP5; Taylor et al., 2012) models, suggested a precipitation decrease across most parts of the MedClim regions (Delworth, et al., 2014, Polade et al., 2014, Mariotti et al., 2015, Araya-Osses et al.,, 2020), with the exception of the northern Mediterranean Basin and northern California, where a precipitation increase during the cold season is expected (Neelin et al., 2013, Polade et al., 2014). However, the model disagreement was found to be high over these parts of the Mediterranean Basin and California, with the models sometimes equally split between suggesting precipitation increase or decline (Polade et al., 2017). Being influenced by both mid-latitude and tropical circulation patterns makes the MedClim precipitation changes more complex to model. It has been suggested that the Hadley cells will expand poleward as a result of global warming, and this is expected to also shift storm tracks polewards and bring drier conditions over some of the MedClim regions (Johanson and Fu 2009, Grise and Davis, 2020). However, the extent to which the Hadley cell will expand is unclear (Chang et al., 2012). On the other hand, the increase of moisture in the atmosphere due to an increase in global temperatures and the Clausius–Clapeyron relation could result in an increase in moisture convergence in some locations, and hence in precipitation increase (Trenberth et al., 2003).

Extreme precipitation events such as floods and droughts are also expected to increase in the MedClim regions (Trenberth 2011; Westra et al., 2013; Trenberth et al., 2014). For instance, precipitation intensity is expected to increase in the future due to the higher concentration of water vapour in the atmosphere, which could intensify wet extremes (rainfall that exceeds the 99th percentile of daily precipitation, for example; Allan and Soden 2008; O'Gorman and Schneider 2009). The Mediterranean Basin is particularly vulnerable to flash floods during extreme rainfall events (Gaume et al., 2016). In addition, more intense droughts are also expected in MedClim regions in the future (Trenberth et al., 2014), together with an increase in the number of dry days (Polade et al., 2014). Future climate projections suggest a decrease in river streamflow across southern Europe, with a decrease in minimum river flows of up to 40 % in the Iberian Peninsula, southern France, Italy and the Balkans (Forzieri et al., 2014; Garcia-Ruiz et al., 2011).

In this study we investigate seasonal and daily precipitation changes in the five MedClim regions as projected by the most recent ensemble of global climate simulations from the Climate Model Intercomparison Project Phase 6 (CMIP6; Eyring et al., 2016), and discuss the model agreement on projected precipitation changes. Given the large uncertainties in future MedClim precipitation projections from the CMIP5 ensemble reported in the previous studies, we study the new CMIP6 simulations and discuss their results, implications and potential improvements. We consider the intermediate ssp2-4.5 and the most extreme ssp5-8.5 scenarios focusing on the future 2050-2079 period, and we use the 1970-1999 as a reference baseline period. The current policies in place on curbing the global greenhouse gas emissions are expected to result in a 2.7°C increase in global mean temperature by the end of the century (Climate Action Tracker, 2022), which would bring us within the range of ssp2-4.5 projected warming for the end of the century. Failure to follow any policies is reflected in the ssp5-8.5 scenario. We analyse monthly and daily precipitation fields, in order to infer the information on mean seasonal changes, shifts in distribution of light, intermediate and heavy precipitation events during the different seasons and changes in the number of



consecutive dry and wet days (CCD and CWD). In Section 2 we describe the data sets and methodology. The results are presented in Section 3, and a discussion and main conclusions are provided in Section 4 and 5.

**2 Methods**

We use data from 25 CMIP6 climate models (see Table S1 for the complete list of models used in the analysis), downloaded from the Copernicus Climate Data Store https://cds.climate.copernicus.eu/#!/home. We study the precipitation differences

between two 30-year periods covering the mid-to-late century (2050-2079) and the historical baseline (1970-1999), under two emission scenarios ssp2-4.5 and ssp5-8.5.

For consistency, monthly and daily precipitation data were regridded to a common 0.7°x 0.7° latitude-longitude grid using bilinear interpolation (CDO command 'remapbil'; Schulzweida 2021). The five MedClim regions were chosen following the Köppen-Geiger climate classification (Kottek, et al., 2006; Peel et al., 2007): the Mediterranean Basin (MED: 29.7°N - 45.5°N,

11.5°W - 35.5°E), California (CAL: 31.5°N - 42.5°N, 125.5°5W - 116.5°W), South America (SAA: 31.1°S - 40.2°S, 75.5°W - 69°W), South Africa (SAF: 31.1°S – 35.3°S, 16.5°E – 20.2°E) and Southwest Australia (AUS: 29°S - 36°5S, 114°E - 129°5E).

The first subsection of the results focuses on seasonal precipitation anomalies, and seasons were defined as winter (Northern Hemisphere (NH): December-January-February; Southern Hemisphere (SH): June-July-August), spring (NH: March-April-

May; SH: September-October-November), summer (NH: June-July-August; SH: December-January-February) and autumn (NH: September-October-November; SH: March-April-May). We also investigate the model agreement on the sign of future changes, showing the percent of models in the CMIP6 ensemble (out of 25 models in total) that agree on either positive or negative sign of future changes in precipitation. In the following subsection, we study percentiles of daily precipitation by season, in order to assess the changes in the distribution of different types of precipitation events in the two future scenarios.

In addition, we investigate the future changes in the consecutive dry days (CDD). We define a dry day when the daily precipitation is ≤ 1 mm day$^{-1}$, and we consider a dry period when we count 5 consecutive days or more of daily precipitation below 1mm day$^{-1}$. Conversely, we study the consecutive wet days (CWD) compared to the historical period. We define a wet day when the daily precipitation is above 5 mm day$^{-1}$, and a wet period as 5 consecutive days or more with daily precipitation above 5 mm day$^{-1}$. Smaller amounts in some of the regions studied can be associated with the strong coastal fogs (Klos et al.,

125 2014).

Finally, we also study extreme precipitation events. For this purpose, we examine the maximum one-day precipitation over a 30-year period, the yearly mean of the maximum one-day precipitation, and the maximum five-day precipitation over a 30-year period for both the historical period and the two future scenarios (ssp2-4.5 and ssp5-8.5).



## 3 Results

In Figure 1 we compare the monthly mean precipitation amounts, spatially averaged over each of the five regions for the 30-year periods considered. During the baseline period (historical simulations spanning 1970-1999), the majority of precipitation occurs in the winter in all five regions, consisting of almost half of the total annual precipitation amount in SAA (49.7%) and CAL (47.6%) and somewhat less in SAF (41.7%), AUS (39.4%) and MED (35.2%). For CAL, the spring share of the total amount is the next most important (27.1%), while in all other regions the autumn contributes more (20-30%).

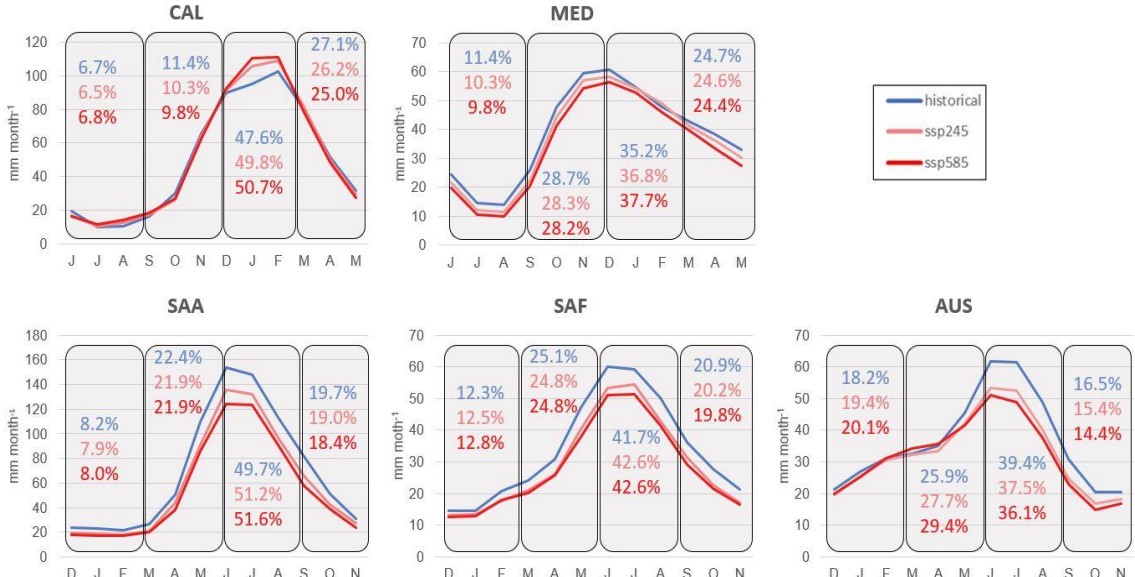


**Figure 1: Seasonal cycle of monthly precipitation (mm month⁻¹) for CMIP6 multi-model ensemble mean for the historical (1970-1999; blue) and future period (2050-2079) under ssp2-4.5 (pink) and ssp5-8.5 (red) scenarios for the five MedClim regions. Shown in percent values are the individual contributions of each season to the total annual precipitation sum.**

In the future, models suggest a decrease in the total annual mean precipitation sum across all regions, except for California

(Table 1). For the ssp5-8.5 scenario, the largest total annual precipitation decrease is projected in SAA (21.6%) and SAF (16.9%) followed by AUS and MED (12.9% and 11.0%). Conversely, annual mean precipitation sum over California will increase by 2.6% under the same scenario. This is due to the winter precipitation increase that is partially compensated by a precipitation decrease in other seasons (Table S3). For the ssp2-4.5, the annual mean cumulative precipitation will decrease by 14.9%, 13.0%, 10.8% and 5.5% for SAA, SAF, AUS and MED regions, respectively. In contrast, CAL shows an increase

in the total annual precipitation under the same scenario by 1.9% (Table 1). In the future, the winter still remains the main contributor to the annual precipitation amount in all regions, but in SAA, SAF and AUS the partitioning of precipitation between spring, winter and autumn becomes more uniform (Table S2 and S3).

Regional precipitation anomalies over California, the Mediterranean Basin and Western Australia are shown in Figures 2-4, and in Figures S7 and S10 for South America (central Chile region) and South Africa, for the ssp5-8.5 scenario. Over CAL,

we find that the winter rainfall will increase compared to the historical period, in coastal California with up to 10% under ssp5-



8.5 scenario (Figure 2a). Polade et al., (2017) and Neelin et al., (2013) report similar results using CMIP5 simulations, and Petrova et al., (2024) comparing CMIP5 and CMIP6 simulations. They found an increase of precipitation during the winter season over central and northern California by 5% to 15 % (Polade et al., 2017). Swain et al., (2018) also found that the projected future winter (November to March) precipitation will increase across most of the state of California, while Lukovic

et al., (2021) indicated that such increase may already be underway.

**Table 2: Annual mean cumulative precipitation in mm year⁻¹ for the five MEDClim regions over the historical (1970-1999) and the future period (2050-2079) under ssp2-4.5 and ssp5-8.5 scenarios. Shown in brackets is the relative change (%) from the historical values.**

|  | Historical | ssp2-4.5 | ssp5-8.5 |
|---|---|---|---|
| CAL | 602.8 mm | 614.3 mm (+1.9%) | 618.3mm (+2.6%) |
| MED | 463.4 mm | 437.9 mm (-5.5%) | 412.3 mm (-11.0%) |
| AUS | 436.5 mm | 389.4 mm (-10.8%) | 380.3 mm (-12.9%) |
| SAA | 835.5 mm | 710.7 mm (-14.9%) | 654.7 mm (-21.6%) |
| SAF | 406.8 mm | 353.9 mm (-13.0%) | 337.9 mm (-16.9%) |

For the summer season, CAL precipitation changes show more regional variability compared to the winter. We find an increase in precipitation of up to 30% in the southeast, while in the northwest part of the region the precipitation decreases by as much as 20% (Fig. 2c). However, it is important to note that summer is generally the dry season in California and that these changes in absolute sense (mm month⁻¹) do not exceed 4 mm (Figure S2c). During the spring and autumn, a decrease of up to 15% and 10% (Fig. 2b and d) is expected in the south and central areas of California respectively.

Future precipitation anomalies in California are however highly uncertain. The intermodel agreement on the sign of future precipitation varies substantially across CAL, as well as seasonally. While during winter, in the north, the model agreement is high (more than 80% of the models agree on the winter season precipitation; Fig. 2e), in the south less than 60% of all models agree on the same sign of future precipitation changes. These findings are similar to the CMIP5 analyses by Polade et al., (2017), Swain et al., (2018) and to the comparison of CMIP5 and CMIP6 by Petrova et al., (2024), suggesting that model

agreement regarding future winter precipitation changes did not improve much in CMIP6 compared to CMIP5 (Petrova et al., 2024). For spring, in the north, the model agreement is low (less than 70%) and high in the south (up to 90%), where we expect drier conditions. During summer and autumn, the intermodel agreement on the sign of future precipitation changes is less than 70% for all of the region, while in the south and inland California models are almost equally split between projecting wetter




and drier conditions with less than 60 % model agreement (Fig. 2g and h). Similar patterns are also found for the ssp2-4.5

scenario, but the intermodel agreement is even lower compared to the ssp5-8.5 (Figure S1).

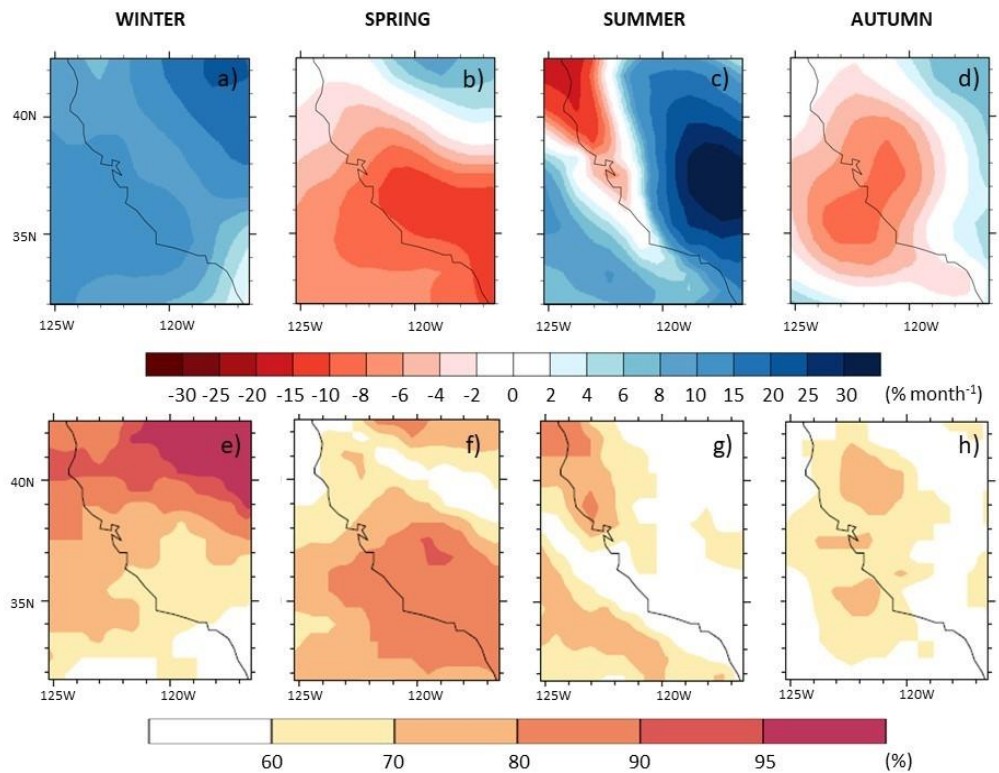

**Figure 2: Future seasonal (winter – DJF, spring - MAM, summer – JJA and autumn – SON) precipitation anomalies expressed in % and model agreement for California (CAL) based on the CMIP6 multi-model ensemble mean changes for mid-late century (2050-2079) ssp5-8.5 future scenario; (a-d): seasonal ensemble mean changes; (e-h): percentage of models agreeing on the same sign of**
**future precipitation changes.**

Across the MED region, mean precipitation changes as well as the intermodel agreement on their sign vary substantially. The

CMIP6 ensemble mean indicates that the Mediterranean precipitation will decrease in all seasons, the only exception being

the most northern parts of the Basin during winter (Figure 3a). An increase in rainfall is seen for the north of Italy (up to 15%),

southeast France (~5%) and northern Balkans (~5%) with a 'no-change' transition zone south of it and winter drying over the

southern parts of the MED region. The northern Mediterranean region is, however, also associated with the highest model

disagreement on the sign of future precipitation changes during winter and spring (Fig. 3 e and f). Thus, the regions with the

low model agreement are the only regions where the ensemble means are not suggesting the precipitation decline. During

summer, most of the MED area exhibits drier conditions, most pronounced in the southwest of the Iberian Peninsula (up to

45%) and the region comprising the Balearic Islands, southern Italy and Malta (up to 30-35%) compared with the reference

period (Fig. 3c). Similar results are also found in previous CMIP5 analyses (Mariotti et al., 2015, Giorgi and Lionello 2008),





that attributed the precipitation decrease to an enhanced anticyclonic circulation and a northward shift of the Atlantic storm tracks.

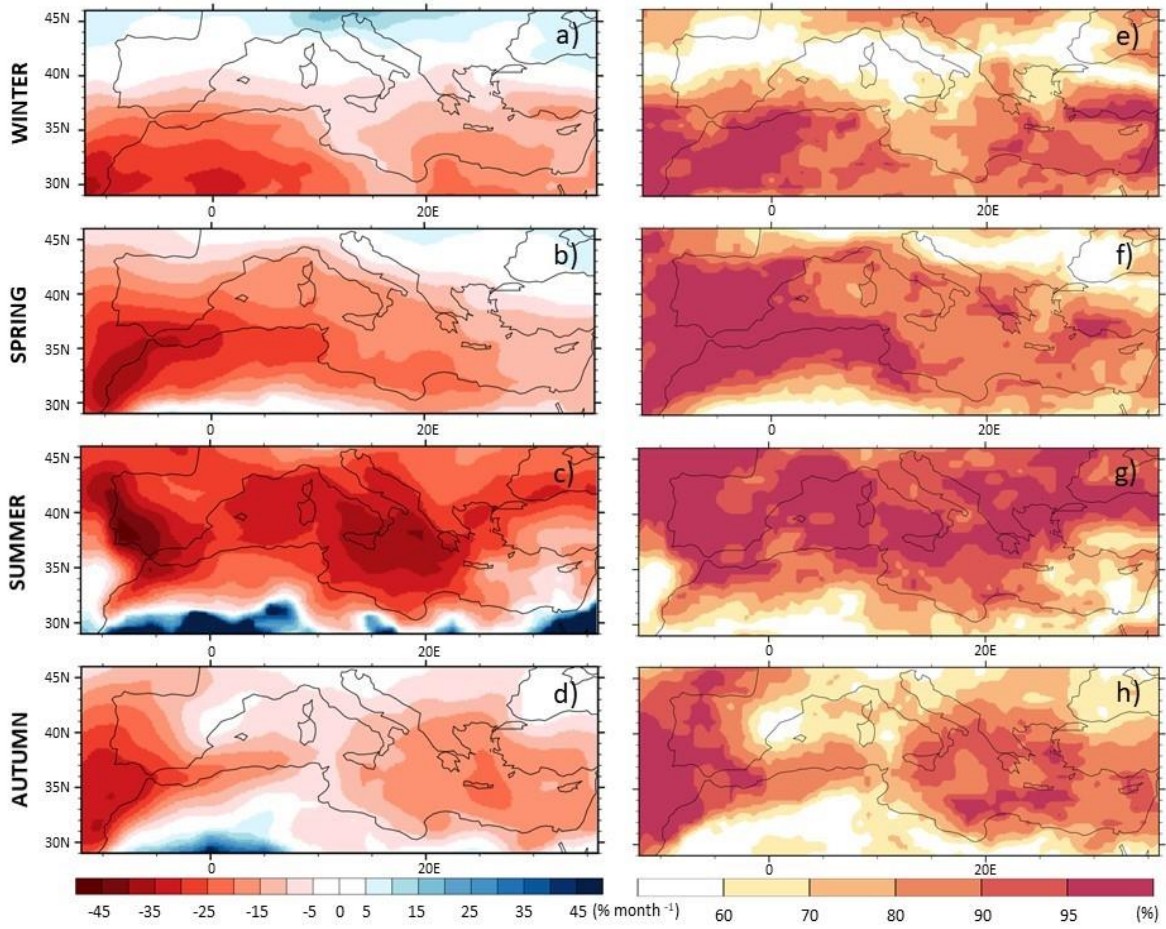

**Figure 3: Future seasonal (winter – DJF, spring - MAM, summer – JJA and autumn – SON) precipitation anomalies expressed in % and model agreement for Mediterranean Basin (MED) based on the CMIP6 multi-model ensemble mean changes for mid-late century (2050-2079) ssp5-8.5 future scenario; (a-d): seasonal ensemble mean changes; (e-h): percentage of models agreeing on the same sign of future precipitation changes.**

During spring and autumn (Fig. 3b and d), the southwestern part of the Mediterranean Basin region is the area expecting the greatest decrease in precipitation, up to 40% (or up to 14 mm per month, see Figure S4b and d). It is worth noting that this is the area where we find the highest model agreement for both seasons (Fig. 3f and h). We also observe similar results for the ssp2-4.5 scenario, but as for the CAL region, this scenario shows smaller magnitude of anomalies and even lower model agreement compared to the ssp5-8.5 scenario (Figure S3).



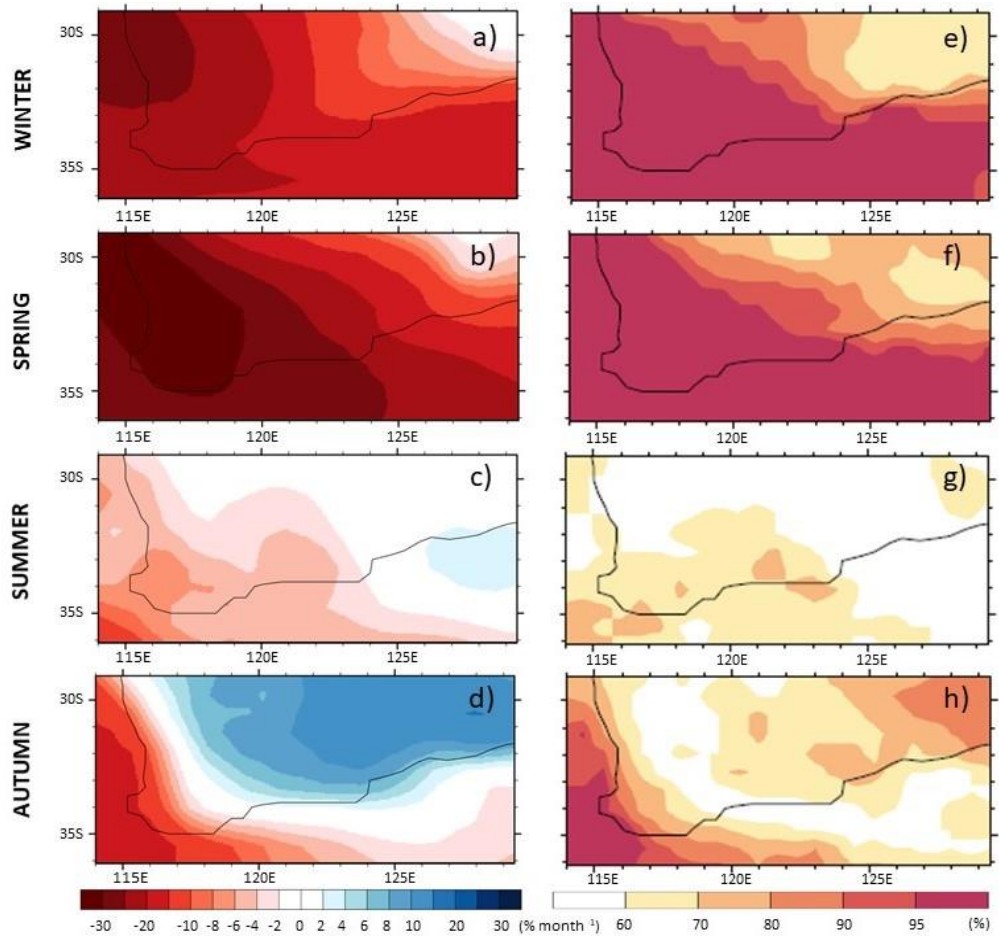

**Figure 4: Future seasonal (winter – JJA, spring - SON, summer – DJF and autumn – MAM) precipitation anomalies expressed in % and model agreement for Southwest Australia (AUS) based on the CMIP6 multi-model ensemble mean changes for mid-late century (2050-2079) ssp5-8.5 future scenario; (a-d): seasonal ensemble mean changes; (e-h): percentage of models (out of 25) agreeing on the same sign of future precipitation changes.**

For the AUS region (Figure 4), precipitation decreases during all seasons in the west, with the most prominent changes in the northwest during the winter and spring seasons. During winter a precipitation decrease of up to 20-25% is seen in the west of the region (Fig. 4a) (or up to 18 mm month$^{-1}$, Figure S6a). Frederiksen et al., (2011) reported similar results using the CMIP3 ensemble, estimating a winter reduction of 10 mm month$^{-1}$ for the period 2040-2059. In the autumn an increase of up to 15% (4 mm month$^{-1}$) is projected in the northern parts of AUS (Fig. 4d). Conversely, in the southwest of AUS, the precipitation is expected to decrease up to 20%. In terms of model agreement, there is a high certainty about the sign of future precipitation changes in winter and spring (Fig. 4e and f), especially in the south of the region (95% of models agree on the drying). However, in summer and autumn (Fig. 4g and h) the model agreement is less than 60% over most of the AUS region. Similar results are found by Andrys et al., (2017), based on CMIP3 models. For the ssp2-4.5 scenario, we find similar results, but the relative precipitation changes as well as the model agreements are lower compared to the ones under the ssp5-8.5 scenario.



During summer, we find a small increase in precipitation in the southeastern part of the region (up to 4%), under the ssp5-8.5 scenario, that is not found in the ssp2-4.5 (Figure S5), but the model agreement is also low.

For SAA and SAF regions precipitation decreases in all seasons (Figure S7 and S10). For SAA the decrease of precipitation occurs mostly in the central parts in summer up to 35% (Figure S7c), and in the northern parts during the other seasons (Figure S7a, b and d). It is important to note that in the winter the largest changes compared to any other season are found in absolute terms, with up to 30 mm month⁻¹ decrease in the central parts of the SAA region (Figure S9a). The model agreement is high compared to other regions, over 95% (Figure S7). Similar findings were reported by Araya-Osses et al., (2020), where at least

a 40% winter precipitation decrease is foreseen under the RCP8.5 scenario in the same region of Chile, and an even greater drying of up to 100% over the summer. The Hadley cell expansion to the south projected under global warming leads to the migration of the South Pacific subtropical anticyclone and storm divergence from the SAA region (Johanson and Fu, 2009), which partially explains these drying patterns. For the ssp2-4.5 scenario, in this region the differences are smaller compared to the reference scenario and the strong model agreement is maintained, with the exception of autumn (Figure S8). Finally, for

SAF, models project drying in all seasons (Figure S10), especially in the northern parts in spring (up to 27%), winter (up to 18%), and autumn (up to 24%). The CMIP5 analysis by Polade et al., (2017) also reported a decrease in the northern parts of SAF in winter of up to 30%, thus greater than the one seen here (Figure S10a). The highest decrease in absolute terms is found in the southwest of the region in winter, up to 12mm month⁻¹ (Figure S12a). Most of the SAF region shows high model agreement, especially for winter and spring, with 95% of models agreeing on the drier future conditions (Figure S12e and f).

The ssp2-4.5 scenario shows a similar pattern as the scenario ssp5-8.5, but with the lower model agreement (Figure S11).

    In Figure 5 we summarise the inter-model differences in projected seasonal precipitation changes based on the regional averages for the ssp5-8.5 scenario. For California, for the winter, models suggest a total range that extends from -8 % to +36% relative precipitation change. Within the 25th-75th percentile range, models suggest wetter winter conditions with values ranging from +1% to +18% (Figure 5a). For the ssp2-4.5 scenario, during winter, the 25-75th percentile range for winter spans

from - 2% to + 17% relative precipitation change (Figure S14a). During spring, models falling in the 25-75 percentile range between -13% to +1% relative precipitation change (Figure 5b). Summer projections exhibit a remarkable range of possible outcomes, spanning from -22 % to almost +80% in precipitation (25-75 percentile range from -10% to +33%; Figure 5c). During autumn, the model projections continue to be spread between wetter and drier future conditions. For autumn, relative precipitation changes from -13% to +8% (the 25-75 percentile range; Figure 5d).

For MED, SAA and SAF regions, the 25-75th percentile range of models suggest drier conditions for all seasons (Figure 5). For the AUS region, models project drier conditions during winter and spring (Fig. 5a and b) and are between wetter and drier outcomes for the other two seasons (Fig. 5c and d). Under the ssp2-4.5 scenario, drier conditions are expected for all seasons over SAA and SAF, but not for the Mediterranean Basin, where winter changes suggest both positive and negative values (Figure S14a).





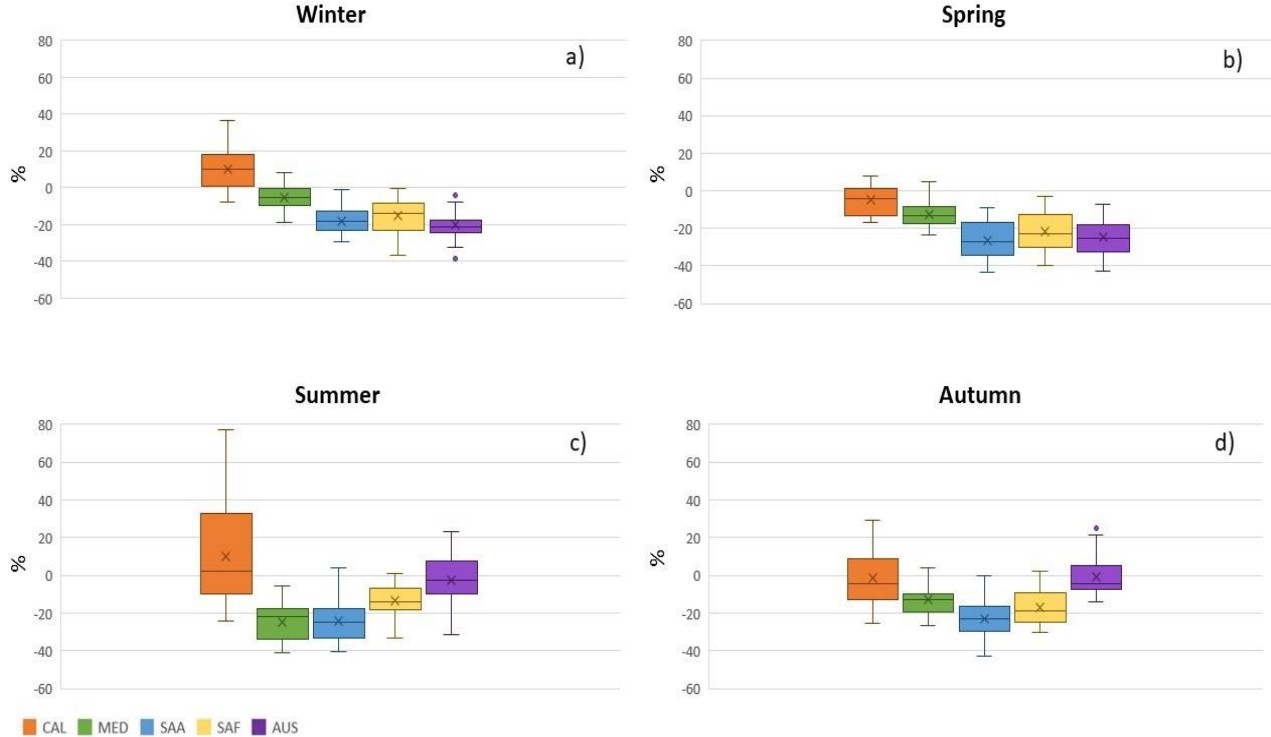


**Figure 5: Box and whiskers plots showing the seasonal precipitation percentage changes for mid-late century (2050-2079) for ssp5-8.5 scenario over the five MedClim regions in the CMIP6 ensemble. Boxes indicate 25 to 75 percentile interquartile range (IQR,) and whiskers minimum and maximum values (values lower (higher) than 1.5xIQR below (above) the 1st (3rd) quartile are shown by dots), ensemble means are indicated by crosses.**

To further illustrate the intermodel differences, in Figure 6 we compare the four most extreme model projections of winter precipitation changes for California and the Mediterranean Basin under the ssp5-8.5 scenario. We select two model projections with the greatest increase in winter precipitation under the ssp5-8.5 scenario, and, conversely, two model projections that show the greatest decrease in precipitation for CAL (Fig. 6a-d) and MED (Fig. 6e-h) regions. The remarkable difference between these projections illustrates the necessity, especially for impact studies, to consider the whole intermodel range of possible

outcomes, and not rely on a single model projection. Likewise, there is a need to develop and utilise physical constraints that would allow subselecting those model simulations that are more likely to have the sign and magnitude of future precipitation changes correct (e.g., through the use of emergent constraints).



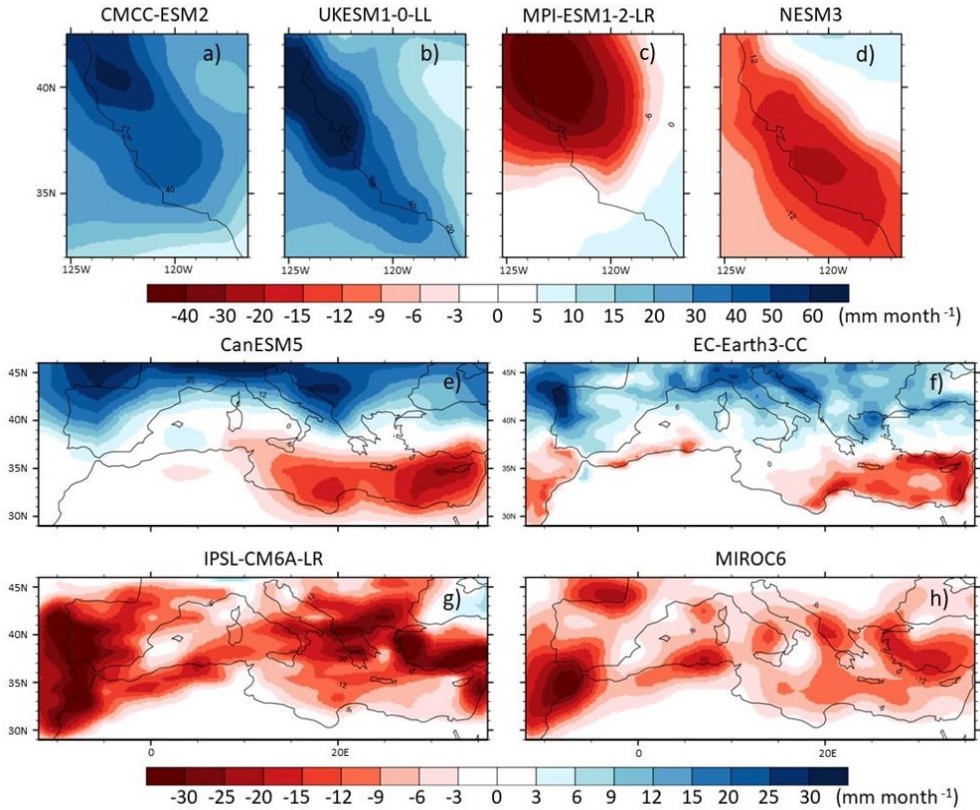

**Figure 6: Seasonal Winter (DJF) precipitation anomalies (mm month⁻¹) from four individual models under the ssp5-8.5 scenario,**
**indicating the most positive and negative models over CAL (a-d) and MED (e-h) regions.**

### 3.1 Daily precipitation changes

In Figure 7, we analyse 1st to 99th rainfall percentiles for each season, to show the differences between ssp5-8.5 scenario and
the historical period. Results for CAL (Fig. 7) show that the intensity of heavy precipitation events (above 90th percentile) will
increase in all seasons. Moreover, during winter we expect an increase in the intensity of all events. Polade et al., (2017) in
their study reported similar results of this winter increase. The precipitation intensity of winter moderate-to-heavy events (70-
89 percentile) will increase by up to 10% compared to the historical period (Figure 7a). Summer also shows an increase in the
intensity of all events, with an increase in moderate (40-49 percentile) precipitation events of 24 % and heavier events (90-99
percentile) of 4% (Fig. 7c). Autumn (Fig. 7d), exhibits an increase in the precipitation intensity of light to moderate (1-49
percentile) as well as heavy (90-99 percentile) events, and a decrease in moderate to heavy ones (50-90 percentile). Finally,
during spring (Fig. 7b) the intensity of all types of precipitation events is expected to decrease with the exception of those
events above 90th percentile. The biggest decrease of nearly 18% is expected in moderate events (40-49 percentile).





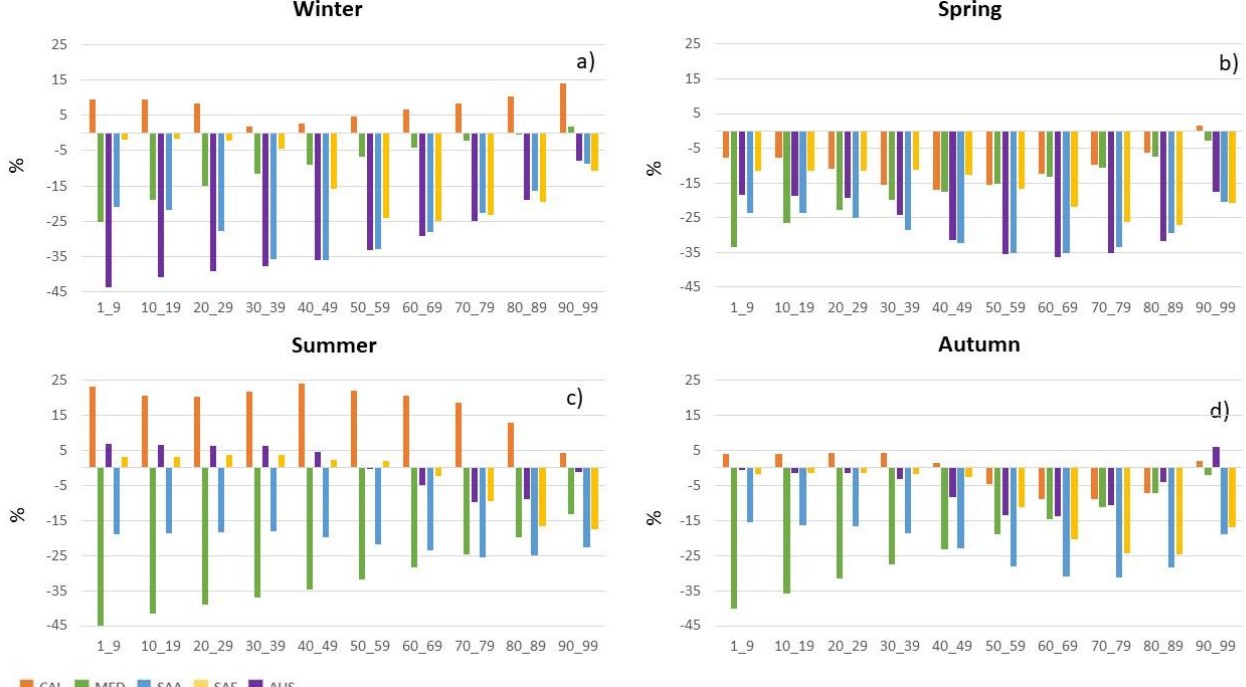

**Figure 7: Precipitation percentiles, 1st to 99th, by range for the five MedClim regions (CAL: California, MED: Mediterranean Basin, SAA: South America, SAF: South Africa and AUS: Australia) for the historical period (1970-1999), and the future (2050-**
**2079) under ssp5-8.5 scenarios for winter (a), spring (b), summer (c) and autumn (d).**

For the MED region (Fig. 7), precipitation events exhibit a decrease in the future, compared to the historical period in all seasons. The greatest decrease is calculated for the light precipitation events. There is, however, a slight increase in the intensity of winter heavy precipitation events (Fig. 7a) in the future of 1.75%. Polade et al., (2017) reported an increase in moderate rainfall events, which we do not find in our study. Summer (Fig. 7c) shows the greatest decreases compared to the historical

period. The decrease for heavy precipitation events above 90th percentile is 13%, for moderate events it is 35%, and for light events below 9th percentile it is 45%. Such drying tendencies in the summer and humid in winter might increase the likelihood of floods and droughts (Alessandri et al., 2014).

In SAA (Fig. 7) the daily precipitation intensity will decrease for all types of events. In winter (Fig. 7a) the largest decrease is projected for the moderate precipitation events between 30 and 59th percentiles of 33 % to 36%, while in spring and summer

(Fig. 7b-c) the largest decrease is between 50th to 89th percentiles of 29% to 35% for spring, and 22% to 25% for summer. For autumn the largest decrease of 31% is between 60th to 79th percentiles. On the other hand, in SAF (Fig. 7), the precipitation intensity of all types of events is expected to decrease in all seasons, with the exception of summer. During the summer (Fig. 7c) light and moderate precipitation events (below 60th percentile) slightly increase up to 3.6%, while events over 60th percentile, exhibit a decrease of up to 18%.

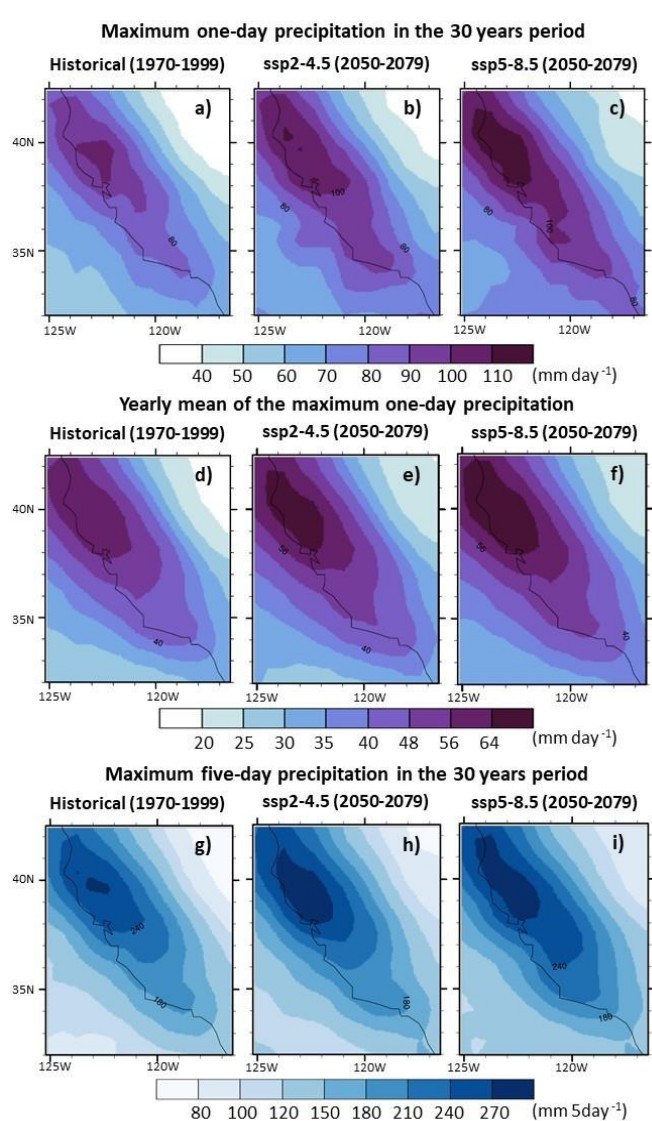


**Figure 8: Maximum precipitation day (mm day⁻¹) over CAL for the 30-year periods: historical (1970-1999) (a) and future (2050-2079) ssp2-4.5 (b) and ssp5-8.5 (c) scenarios. And ensemble mean of the maximum precipitation day (mm day⁻¹) in every year over the 30-year period for historical (d) and future ssp2-4.5 (e) and ssp5-8.5 (f) scenarios. And maximum precipitation during 5 days (mm 5day⁻¹) for the 30-year periods: historical (g) and future ssp2-4.5 (h) and ssp5-8.5 (i) scenarios.**

Finally, in AUS (Fig. 7) precipitation intensity is expected to decrease in all seasons, except for the summer (Fig. 7c) in which

the intensity of the events below the 50th percentile is expected to increase up to 7%. The moderate and heavy events (above

percentile 50th) are expected to generally decrease by up to 10 %. There is, however, a slight increase in the heavy precipitation

events above 90th percentile by 6 % in autumn (Fig. 7d), which might indicate increased probability for extreme precipitation

events. Similar results are observed for the ssp2-4.5 scenario for all the regions and seasons (Figure S16). Our results suggest



an overall drying tendency in the MedClim regions, accompanied with an increase of heavy precipitation events in California, during winter and Southwest of Australia in autumn, based on the ssp5-8.5 scenario.

Maximum one-day and five-day precipitation and the annual mean of the maximum one-day precipitation for the historical and future periods are shown in Figure 8. The CAL region exhibits the largest increase compared to other MedClim regions. The north-west of CAL will experience a maximum one-day precipitation of up to 110 mm per day (Figure 8c), which is 10 mm day$^{-1}$ more than for the historical period (Fig. 8a). The annual mean of the maximum one-day precipitation is also projected to increase, up to 64 mm day$^{-1}$ for mid-late century (Fig. 8d-f), 8 mm day$^{-1}$ more compared with the historical period. We further show that the wettest parts of California, with the maximum amount of precipitation over five consecutive days of 270 mm, are projected to spatially expand in the future (Fig. 8g-i). Pierce et al., (2013) and Polade et al., (2014) found an increase in the number of heavy precipitation days in the northern part of California. Huang et al., (2020) reported a substantial increase in total event precipitation accumulations over the entire state under the extreme emissions scenario (2071-2080 under RCP8.5 scenario). Moreover, they found that there will be a substantial increase in intense atmospheric rivers resulting in such precipitation extremes in the future.

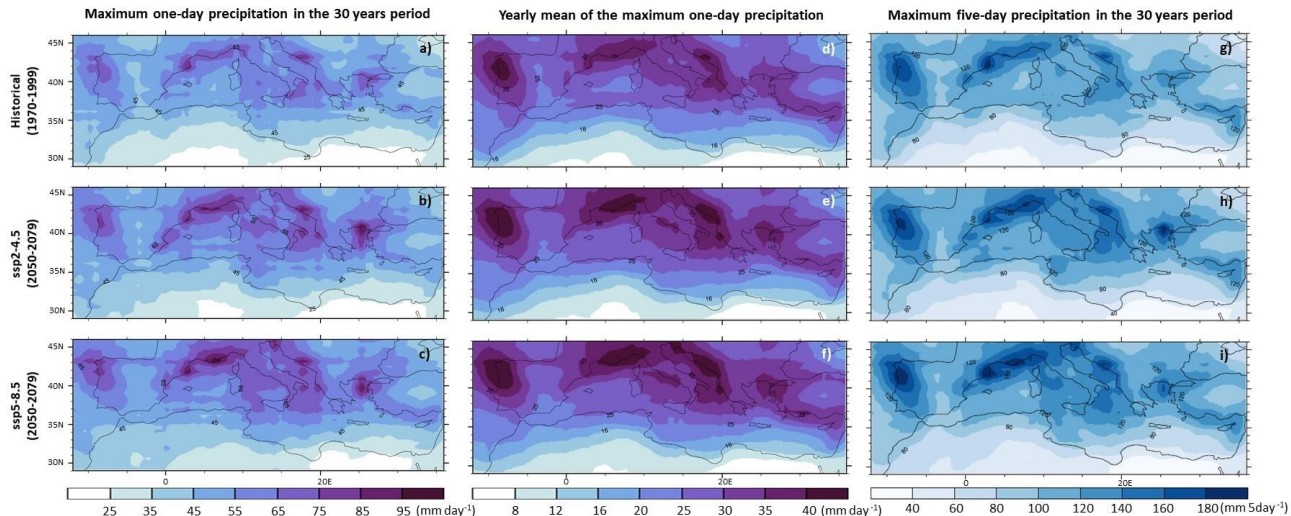

**Figure 9: Maximum precipitation day (mm day$^{-1}$) over MED for the 30-year periods: historical (1970-1999) (a) and future (2050-2079) ssp2-4.5 (b) and ssp5-8.5 (c) scenarios. And ensemble mean of the maximum precipitation day (mm day$^{-1}$) in every year over the 30-year period for historical (d) and future ssp2-4.5 (e) and ssp5-8.5 (f) scenarios. And maximum precipitation during 5 days (mm 5day$^{-1}$) for the 30-year periods: historical (g) and future ssp2-4.5 (h) and ssp5-8.5 (i) scenarios.**

In the MED region, the areas with the highest daily rainfall are located in the northwest of the Iberian Peninsula, south-east of France and the eastern coast of the Adriatic Sea. All three variables analysed are projected to increase for mid-late century, possibly as a result of the enhanced evaporation (Toreti et al., 2013). The wettest parts of the region are projected to experience maximum one-day precipitation of 95 mm day$^{-1}$ (Figure 9a-c), as well as maximum five-day precipitation up to 180 mm (Figure 9g-i). Such an increase of extreme precipitation intensity in the MED (Larsen et al., 2009; Hosseinzadehtalaei et al., 2020), can pose problems for urban drainage systems thus increasing the risk of potential flooding.



The areas with a maximum one-day precipitation for the AUS region are located in the north (Figure 10), reaching 95 mm
day⁻¹ in the future (in both scenarios). The annual mean maximum one-day precipitation range is very similar to the other
MedClim regions. Such an increase in daily precipitation intensity, and heavy precipitation for the remainder of the 21st
century has been previously reported by Alexander and Arblaster (2017).

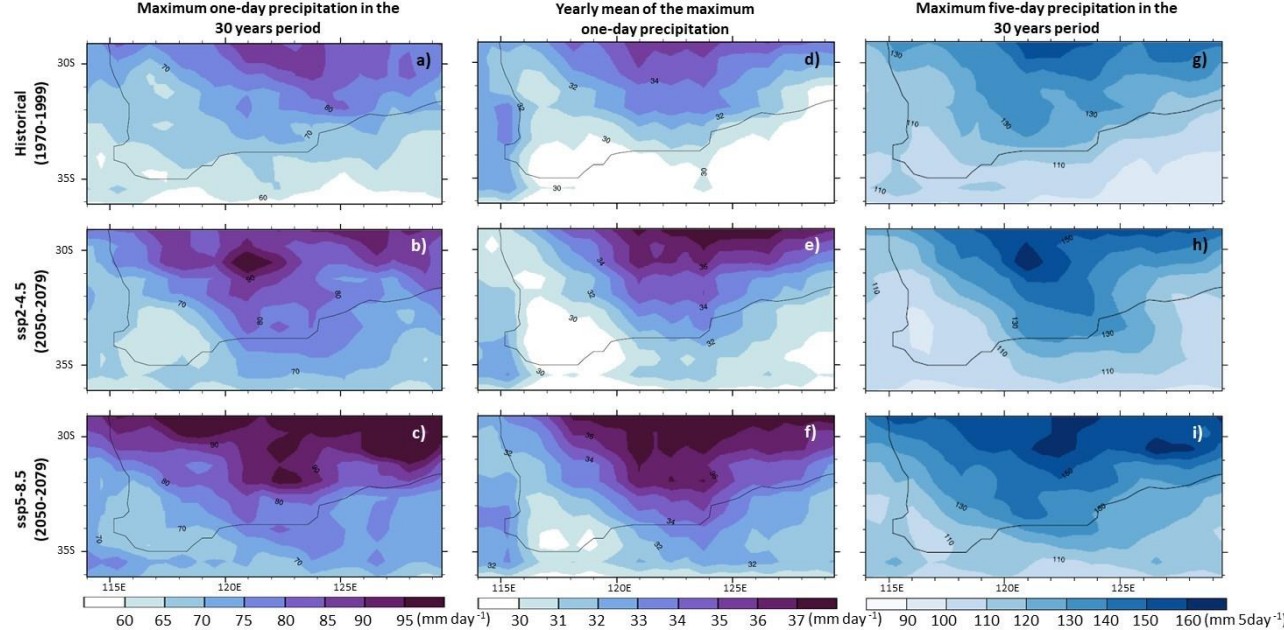

**Figure 10: Maximum precipitation day (mm day⁻¹) over AUS for the 30-year periods: historical (1970-1999) (a) and future (2050-**
**2079) ssp2-4.5 (b) and ssp5-8.5 (c) scenarios. And ensemble mean of the maximum precipitation day (mm day⁻¹) in every year over**
**the 30-year period for historical (d) and future ssp2-4.5 (e) and ssp5-8.5 (f) scenarios. And maximum precipitation during 5 days**
**(mm 5day⁻¹) for the 30-year periods: historical (g) and future ssp2-4.5 (h) and ssp5-8.5 (i) scenarios.**

The maximum one-day precipitation in SAA is projected first to decrease under ssp2-4.5 (Figure S17b), and then slightly
increase under the ssp5-8.5 (Figure S17c). In contrast to the other regions, the annual mean of maximum one-day precipitation
is projected to decrease in SAA for mid-late century by up to 5 mm day⁻¹ (Figure S17e-f). Although our results for SAA
suggest a decrease of maximum one-day precipitation, Hodnebrog et al., 2022 report a significant variability between different
model simulations of the maximum one-day precipitation in Chile for the 21st century.

In SAF the area with the highest maximum one-day, and five-day precipitation in the 30-year period, as well as the yearly
mean of the maximum one-day precipitation is located in the south, 70mm day⁻¹ 130 mm 5day⁻¹, respectively (Figure S18).
Using CMIP5, Pohl et al., (2017) obtain similar results, but we find slight changes for the highest-intensity precipitation days
in the future. In contrast, Abiodun et al., (2020), using CMIP5, has found an increase of more intense rainfall events in this
region. Therefore, we assume that such differences could be possibly explained by the use of different CMIP generations of
models.



## 3.2 Changes in Consecutive Dry and Wet Days

Changes in Consecutive Dry Days (CDD) and Consecutive Wet Days (CWD) for the two future scenarios for CAL, MED, and AUS are shown in Figures 11-13, respectively. The southern parts of CAL in the historical period have the greatest number of CDD with more than 190 days year⁻¹ with less than 5 mm day⁻¹ of precipitation, as opposed to the north, where some areas do not exceed 90 days year⁻¹ (Fig, 11a). In the future, we observe a west-east spatial dipole of CDD increase along the coast, and decrease inland, along the Sierra Nevada Mountains, both by up to 8 days year⁻¹, for ssp5-8.5 scenario (Figure 11c). Although

Polade et al., (2017) show similar results for the winter, some authors argue that extremely dry years are due to a rise in dry days in general, not specifically CDD (Cayan et al., 2010, Diffenbaugh et al., 2015, Swain et al., 2018). The number of CWD barely changes under the two different scenarios (Fig. 11 g-h). Regarding the intermodel agreement on the projected anomalies of CDD and CWD, we found weak model consensus. CAL generally shows less model agreement compared to the other MedClim regions (Fig. 11d-e, 11i-j).

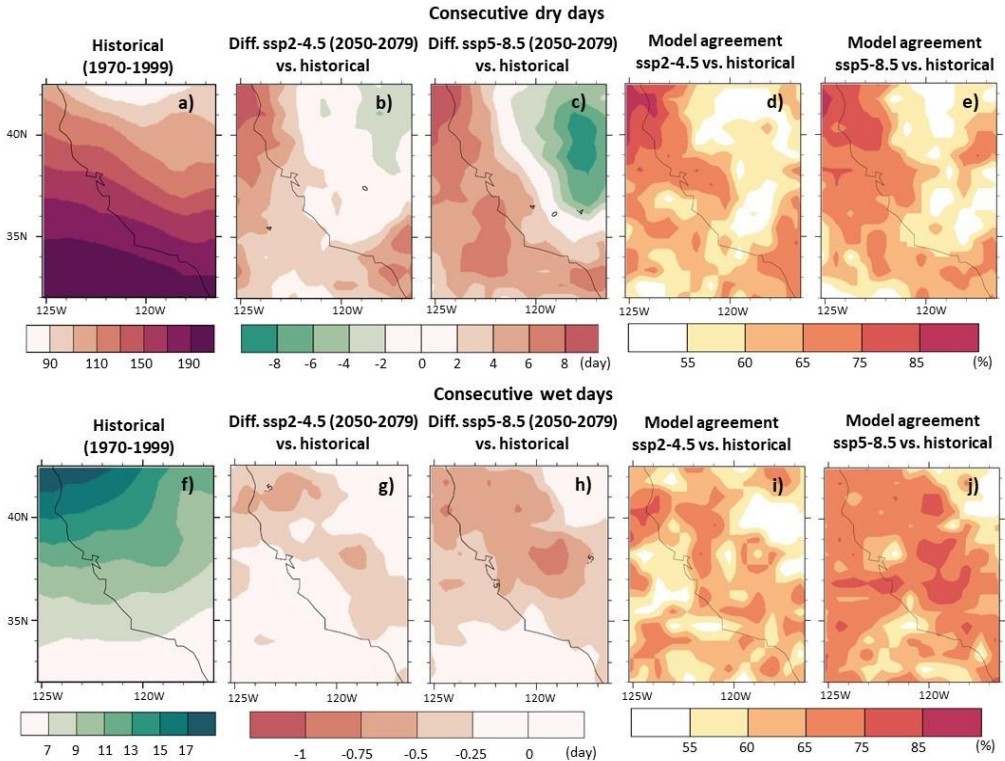


**Figure 11: Ensemble mean of annual consecutive dry days (CDD) for historical period (1970-1999) (a) and differences between historical and mid-late century (2050-2079) for ssp2-4.5 (b) and ssp5-8.5 (c) scenarios. And model agreement in percentage on the same sign of future precipitation changes for ssp2-4.5 (d) and ssp5-8.5 (e) scenarios over California (CAL). Same for CWD represented from (f-j).**

In the MED region the greatest increase in CDD of up to 22 days is expected in the maritime and coastal areas, especially in its southwestern parts (Fig. 12b-c). These results are in agreement with a recently published study by Todaro et al., (2022)





which found an increase in the number of CDD in the future for the scenario RCP8.5. Interestingly, we find that the areas close to the Sahara Desert will barely experience changes in CDD (Figure 12), which is the opposite to the findings by Lionello and Scaracia (2020), who reported a much larger CDD increase in these parts using CMIP5 models. For the CWD, a slight

decrease is expected in the future (Figure 12g and h). Our results confirm previous findings by Dubrovsky et al., (2014) who reported an overall decrease in the probability of wet days for the MED region. The model agreement for the MED region demonstrates higher consensus compared to CAL. The areas experiencing the greatest increase in CDD also exhibit high model agreement (95%) (Fig. 12d and e). Conversely, the results for CWD display lower model agreement compared to CDD, with reduced uncertainty within the Mediterranean basin itself (Fig. 12i and j).

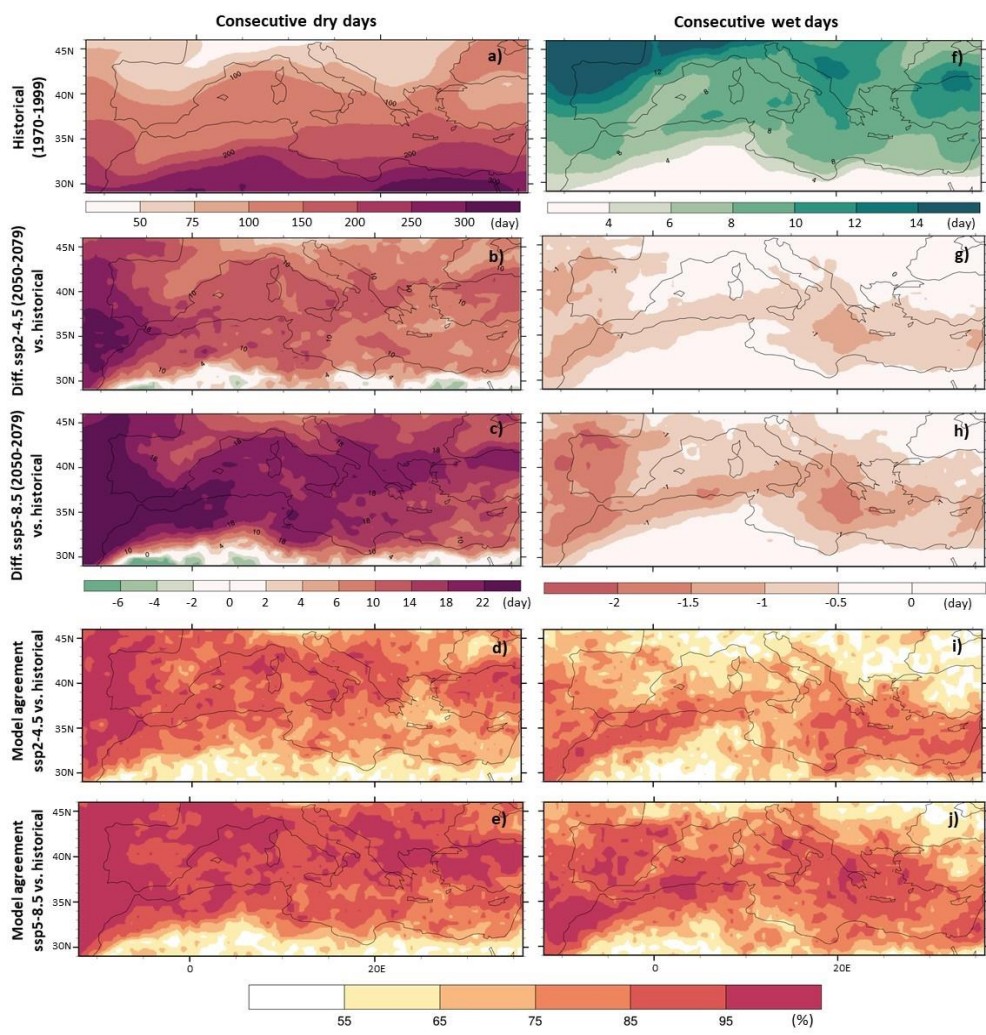


**Figure 12: Ensemble mean of annual consecutive dry days (CDD) for historical period (1970-1999) (a) and differences between historical and mid-late century (2050-2079) for ssp2-4.5 (b) and ssp5-8.5 (c) scenarios. And model agreement in percentage on the same sign of future precipitation changes for ssp2-4.5 (d) and ssp5-8.5 (e) scenarios over Mediterranean Basin (MED). Same for CWD represented from (f-j).**




The central parts of AUS, already associated with 125 CDD per year (Figure 13a), will further experience an increase of 14 CDD per year under the ssp5-8.5 scenario (Fig. 13b and c). For CWD a slight decrease of up to 3 days is expected under the ssp5-8.5 scenario (Fig. 13g-h). These results generally agree with those of Andrys et al., (2017), who also found an increase in CDD and a decrease of CWD for mid-century using CMIP3. For this region, CWD exhibits high model agreement (95%),

particularly in the southern parts (Fig. 13i and j).

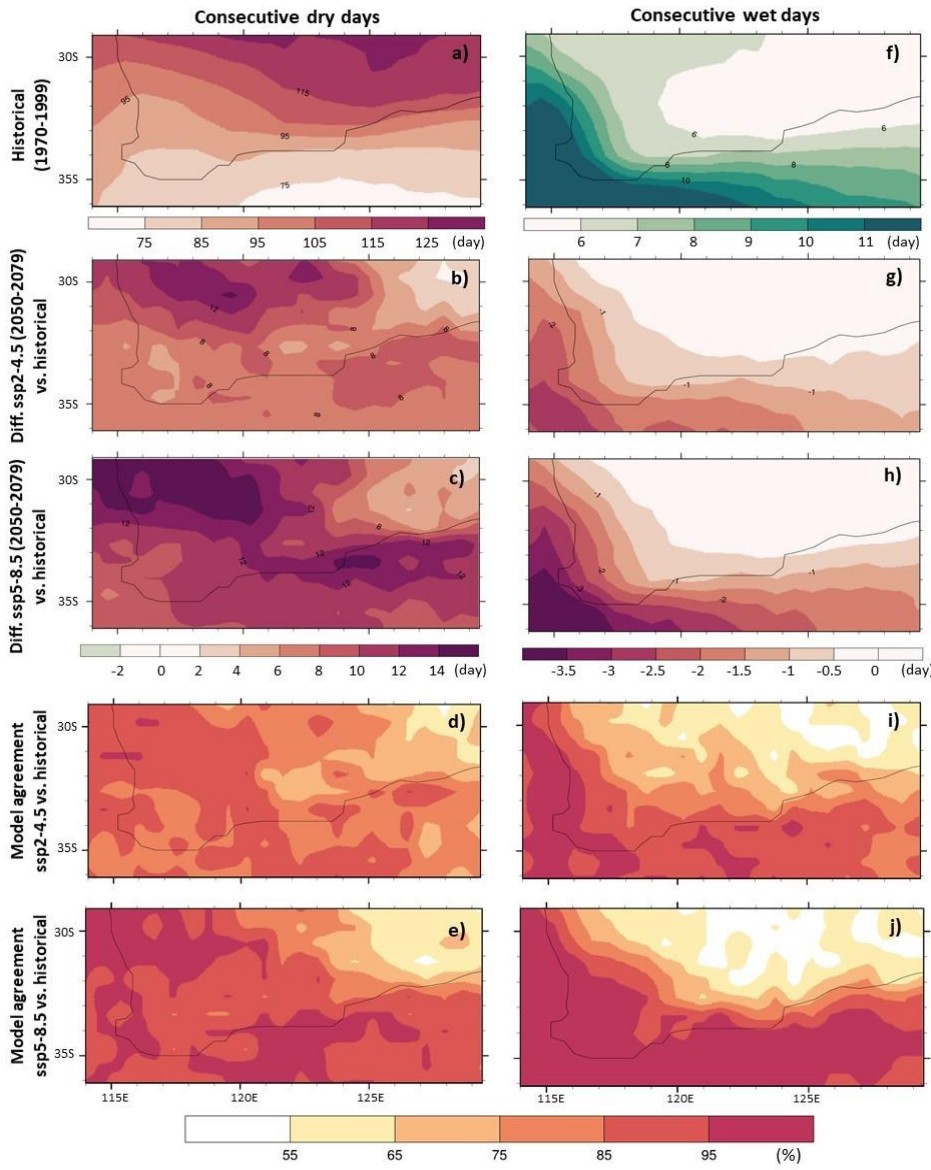

**Figure 13: Ensemble mean of annual consecutive dry days (CDD) for historical period (1970-1999) (a) and differences between historical and mid-late century (2050-2079) for ssp2-4.5 (b) and ssp5-8.5 (c) scenarios. And model agreement in percentage on the**



**same sign of future precipitation changes (d-e) for ssp2-4.5 (d) and ssp5-8.5 (e) scenarios over Australia (AUS). Same for CWD represented from (f-j).**

Lastly, the SAA and SAF regions are shown in the Supplementary Material (Figures S19 and S20). They both have the highest number of CDD in the northernmost parts, with an expected increase of up to 21 days. In SAA, the greatest increase is expected in the coastal areas (Figures S19b-c), while in SAF the greatest increase will occur inland (Figures S20b-c). A considerable decrease in CWD is expected in the southern parts of SAA, up to 3.5 days under the ssp5-8.5 scenario (Figure S19h). Both

regions show a high model agreement for the ssp5-8.5 and ssp2-4.5 scenarios (Figures S19 and S20, panels d, e, i and j).

## 4 Discussion

Water scarcity has become a pressing issue for many regions of the world due to an increase in water use and the changing climatic conditions (WWAP, 2019). Since the 1980's, water use has been increasing worldwide by about 1 % per year, driven by a combination of population growth, socio-economic development and changing consumption patterns (WWAP, 2019).

Compared to 50 years ago, approximately 163 million more people globally live under unfamiliar dry conditions due to a changing climate (Pörtner et al., 2022). By 2050, global water demand is expected to increase by 20-30% above the current water use, mainly as a consequence of rising demand in the industrial and domestic sectors (WWAP, 2019). Historically, Mediterranean climate regions, such as California, the Mediterranean Basin and South Africa, have faced water scarcity issues, but the recent exceptional droughts (e.g. 2011-2014 California drought, 2021-2023 western Mediterranean drought, 2015-2018

Cape Town drought; Seager et al., 2015; Toreti et al., 2023c; Parks et al., 2019) have challenged and triggered changes in water management planning and policies (Berbel and Esteban, 2019). These droughts are posing an increasing threat for agriculture, industry and health (Cramer et al, 2018). Our analysis of CMIP6 projections suggests that in the future the annual precipitation sum will decrease over all MedClim regions except for California that will see a small increase (Table 1). The precipitation may increase during some individual seasons; model projections suggest an increase over the north of the

Mediterranean Basin in winter (Figure 3a), southwest Australia in autumn (Figure 4d) and southern California during winter and summer (Figure 2a and c). However, in these seasons and regions the model agreement on the sign of future precipitation changes is very low, giving little certainty about the projected wetter conditions. In general, the model agreement on the sign of future precipitation changes is high over the regions and seasons where the ensemble mean indicates precipitation decline in the future and low elsewhere. In other words, for MedClim regions except California, models provide certainty only about

the conditions getting drier.  Next, we discuss the implications of our analysis for each MedClim region separately.

*California*

Eighty percent of California's water demand comes from the southern two-thirds of the state (California Department of Water Resources, 2023).  The Sierra Nevada mountains are a crucial water source for California, primarily through the gradual melting of snowpack, which feeds various lakes and rivers, ultimately flowing into the Sacramento and San Joaquin rivers

(Klausmeyer and Fitzgerald, 2012). These rivers, in turn, provide water to the Sacramento-San Joaquin Delta, which serves



approximately 27 million people through the State Water Project developed in 1962 (California Department of Water Resources, 2023). Additionally, many cities in southern California get some of their drinking water from the Colorado River, which is diverted into the Metropolitan Water District's Colorado River Aqueduct, transporting the water 242 miles to southern California (Klausmeyer and Fitzgerald, 2012).

We find that the Sierra Nevada region will have more rainfall during the winter months in the future (10% increase under ssp5-8.5 and 4% increase under ssp2-4.5) (Figure 2a and Figure S1a) with a modest model agreement of about 70% for the ssp5-8.5 scenario (Figure 2e) and low model agreement of about 60% for ssp2-4.5 scenario (Figure S1e). In spring and autumn however, this region will experience a precipitation decrease of 8% for both seasons in ssp5-8.5, and 4% (spring) and 6% (autumn) under ssp2-4.5 (Figure 2b and d and Figure S1b and d). In terms of Consecutive Dry Days, the Sierra Nevada region

will see no change and southern California will see an increase of 6 days year$^{-1}$ under ssp5-8.5 and 4 days year$^{-1}$ under ssp2-4.5 (Figure 7).

The southern region of the state of California, which gets some of their drinking water from the Colorado River, will experience precipitation increase in summer, winter and autumn (low model agreement). In spring a decrease of ~ 10% is expected under the ssp5-8.5 scenario (modest model agreement of 80%) (Figure 2). For the ssp2-4.5 scenario, precipitation projections show

low model agreement for all seasons (~ 60%), with winter showing a small increase of 2%, and in spring and autumn a small decrease (Figure S1).

Although our results indicate that northern California is expected to experience an increase in precipitation, the reservoirs of this region are centralised, leading to a limited capacity to adapt to precipitation changes (Public Policy Institute of California, PPIC 2018). On the other hand, our results show substantial uncertainties in projections of future precipitation changes across

southern California, which present a challenge for water supply planning and management. Southern California has a long tradition of dealing with water scarcity and high water demand from the agriculture sector (Quandt et al., 2022). Our results show uncertainty in the model projections that will present a major challenge on water management in this area. In addition, a decrease in water availability from the Colorado River, which lies between the California and the Arizona border, could create tensions between these states. Clearly, water storage and distribution planning is a key issue for the state of California, and an

emergency drought response is necessary to ensure water availability in this state.

*Mediterranean region*

The water resources in the Mediterranean region are unevenly distributed, with the majority of resources (>70%) located in the north, 23% in the east, and only ~5 % in the south (Plan Bleu report, 2005). This north-south gradient in water availability represents a major challenge for water management. For example, in the Iberian Peninsula the two main sources of surface

water are the Pyrenees that feed the Ebro basin in the north-east and the Tagus basin in the west. These two basins provide water to the rest of the region through interbasin water transfers (Ministry of the Environment, 2003). In Italy, almost 53% of the usable surface water resources are located in the north of the country, Po River Basin and Alps (Ministry for the Environment, Land and Sea, 2005). Morocco also follows a sharp gradient in north-south water availability, with the northern basins (Loukkos, Tangérois and Coastal Mediterranean) and Sebou basin, which have more than half of the water resources



but cover only around 7 per cent of the country's surface area (United Nations Economic Commission, 2022). Finally, in the east of the Mediterranean basin, in northern Turkey, Omerli Reservoir is one of the major water reservoirs of Greater Metropolis Istanbul, providing 40% of the overall water demand (Coskun and Alparslan, 2009).

The severity and frequency of droughts has increased in Europe, there has been a prolonged drought from 2021 up until today (European Commission, Joint Research Center, 2023). Droughts are causing economic loss in Europe, and the countries in the
Mediterranean basin are the most affected. Relative to the size of the economy, present drought losses are highest in the Mediterranean (on average 0.12% of GDP), for the period 1990-2016, annual losses per year from drought are around 1.5 billion € in Spain, 1.4 billion € in Italy and 1.2 billion € in France (Cammalleri et al., 2020). Farmers have been adversely affected through the damaging of crops and reduction in production, which has led to economic losses. This may cause a further increase in their costs due to difficulties in accessing water, for instance through increasing water prices or the drilling
of new wells (Bisselink et al., 2020).

Spain is the leading producer and exporter of fruits and vegetables in the EU region, considered "the vegetable garden of Europe", accounting for over 26% of European production and ranking seventh globally (Ministry of Agriculture, Fisheries and Food, 2023). This export plays a crucial role for the Spanish economy, with a value exceeding 18 billion euros (Ministry of Agriculture, Fisheries and Food, 2023). The biggest share of total irrigated agriculture is found in the south-east of Spain,
which has been particularly affected, for instance reservoirs have reduced their capacity from 40% to 25% (Toreti et al., 2023b). In 2001, Spain developed its National Hydrological Plan, focusing on distributing water from the north to the south-east regions (Ministry for Ecological Transition and Demographic Challenge, 2023). However, water transfers to the south-east region were limited in 2022, showing that this water management strategy may no longer be sustainable, due to the decrease in water availability in the Tagus region.

In Italy, the recent drought has seen the water levels in most reservoirs fall below the minimum historical values (1970-2019; Toreti et al., 2022a). For instance, the water deficit in the Italian Alps compared to the median conditions from 2009 to 2021 was recorded at -61% (Toreti et al., 2022b). Furthermore, in July 2022 the Italian government declared a state of emergency due to drought in five regions. Water restrictions were implemented in hundreds of municipalities, particularly in the northern part of the country, primarily affecting the Po River basin. In 2023, various sectors were affected by the lack of water, and in
the Po River basin, the production of hydroelectric power was reduced, affecting the power market, while the agricultural sector experienced a loss of 26000 ha of rice paddies (Toreti, 2023b).

Furthermore, in other Mediterranean countries located in the North of Africa, such as Morocco and Tunisia, drought has led to a reduction in cereal agricultural production in 2023, due to the lack of precipitation since October 2022. In Turkey, the warm winter of 2023 reduced snow accumulation, raising concerns about the lack of snow contribution to the water flow. Due
to this drought, in February 2023, reservoirs were below 50% of their capacity. Additionally, among the enormous impacts of the catastrophic earthquakes that hit Turkey and Syria in February 2023, there were damages to dams and river rendering reservoirs unusable for storing water for the dry season (Toreti et al., 2023a).



In the future, we expect precipitation in the MED region to decrease during all seasons and regions, except in winter over the south-east of France, north of Italy and the eastern coast of the Adriatic Sea, for both scenarios (Figure 3). In particular, the

south of the Iberian Peninsula, south of Italy, Greece and Turkey are the areas where it is expected to be drier in the future all year round.

In Spain, the Pyrenees region is an important water source for the whole country, but climate model projections suggest drier conditions for all seasons except during winter, where models are split between wetter and drier conditions, with the latter meaning a real threat to water supply. The model agreement during winter in the Pyrenees region is low (less than 60% for

both scenarios; Figure 3e and Figure S3e), and models do not project any changes during this season for both scenarios (Figure 3a and Figure S3a). In addition, the Tagus region, located in the western Iberian Peninsula, also constitutes an important water reservoir, and it is expected that there will be a decrease in precipitation in this area, especially during summer (up to 45% for ssp 5-8.5 and up to 35% for ssp2-4.5 scenarios; Figure 3c and Figure S3c). In the Tagus region, during summer, models show high model agreement (95% for both scenarios; Figure 3g and Figure S3g). On the other hand, Consecutive Dry Days will

increase in both regions in the future. For the Pyrenees region, the increase in CDD will be 10-14 days year$^{-1}$ for the ssp5-8.5 scenario (Figure 8c) and 6 days year$^{-1}$ for the ssp2-4.5 scenario (Figure 8b). In the Tagus region, the increase of CDD will be 22 days and 14-18 days for the ssp5-8.5 and ssp2-4.5 scenarios, respectively (Figure 8c and b). These two regions do not project changes in extreme precipitation events (Figure 11). These results show that water availability in Spain in the future could become a greater problem through an increase in the duration and intensity of droughts.

In Italy, the Alps represent an important water reservoir for the whole country, but our simulations indicate that precipitation in this area in the future will decrease in all seasons except in winter, where we found an increase of up to 15% for both scenarios (Figure 3a and Figure S3a), and with a relatively high model agreement (70-80%; Figure 3e). The CDD will increase in this region up to 10 days year$^{-1}$ for the scenario ssp5-8.5 and up to 6 days year$^{-1}$ for the ssp2-4.5 scenario (Figure 8c and b respectively). Similarly to Spain, in northern Italy, extreme precipitation events will not change in the future (Figure 11). The

south (Apulia, Basilicata, Sicily and Sardinia) already experiences water shortages, and model projections suggest that this will worsen in future, especially in summer, where precipitation will decrease up to 35% for the scenario ssp5-8.5 (Figure 3c) and 30% for the scenario ssp2-4.5 (Figure S3c) with a high model agreement (90-95% for ssp5-8.5 in Figure 3g and 80% in Figure S3g). In the south of Italy, CDD will increase up to 18 and 10 for ssp5-8.5 and ssp2-4.5 scenarios respectively, leading to a greater risk of drought in the future.

The wettest areas in the north of Africa and Turkey are found in the north, and in these regions model projections suggest drier conditions for all seasons, particularly in summer. In the Black Sea region of Turkey, models show high agreement during the summer (95% for both scenarios; Figure 3g and Figure S3g). Conversely, in North Africa, high model agreement is observed during spring (95% for both scenarios; Figure 3f and Figure S3f). Furthermore, the number of Consecutive Dry Days is projected to increase in both regions in the future, reaching up to 22 days. While the northern regions of Africa do not indicate

changes in extreme precipitation events, the coastal areas of Turkey are expected to experience an increase in extreme precipitation events (Figure 11).



Currently, droughts already pose a challenge for Europe, especially for the Mediterranean Basin. For this reason, it is of paramount importance to develop adaptation plans tailored to countries most vulnerable to drought, such as Spain and Italy. As our results show, precipitation in the Mediterranean basin is expected to decrease overall. These conditions could potentially

lead to an increase in the severity of droughts in Europe in the future and increase in economic costs. Due to the effects of climate change only, in the Mediterranean and Atlantic sub-regions, annual drought related damages in the agriculture sector will double from 3.3 €billion per year in the baseline to 8 € billion per year at 3°C of warming (Cammalleri, et al., 2020). Moreover, drought also limits hydroelectric energy production, damage in this sector in these regions will increase from 1.4 to 3.3 € billion per year (Cammalleri, et al., 2020).

*Southwest Australia*

The largest reservoir of the southwest region is the Wellington reservoir, found in the Collie river (Government of Western Australia, 2023). Further north, the Perth area is notably surrounded by dams, which traditionally relied heavily on streamflow, but in recent years streamflow has declined leaving them depleted (Water Corporation, 2023).

For both regions, projections suggest drier conditions for all seasons, especially in spring (30% for ssp5-8.5 and 20% for ssp2-

4.5; Figure 4b and Figure S5b). The model agreement during spring is very high (95% for both scenarios; Figure 4f and Figure 5f). Consecutive Dry Days will increase in the future in the Wellington reservoir and Perth, 8days year$^{-1}$ for ssp5-8.5 and 4days year$^{-1}$ for ssp2-4.5 (Figure 9c and b). Model projections do not show changes in extreme precipitation events over these two regions (Figure 12). Frederiksen and Osbrough (2022) warned that a dramatic rainfall reduction could result in reservoirs depletion, for example the Perth reservoir total capacity was reduced by 20% during the last decade as compared to before the

1960s. The Integrated Water Supply Scheme can no longer rely solely on rainfall, and now includes desalinated seawater and groundwater replenishment (Water Corporation, 2023).

In Southwest Australia, climate change has already dramatically affected the water resources, and it is expected to reduce stream flow into dams and groundwater recharge in the western and southwestern areas (Department of Primary Industries and Regional Development, 2021). Our results confirm this threat, indicating a very high certainty about the conditions becoming

drier.

*South America*

In central Chile (SAA), the subtropical Andes act as the main water reservoir for the region, as the winter precipitation contributes to the seasonal snow cover, which melts in spring and summer to supply river flow (Center for Climate and Resilience Research, 2023). According to our results, this region is expected to become vulnerable to droughts in the future,

with a precipitation decrease projected during all seasons (Figure S7-9). Winter projections suggest a decrease in precipitation of 15% for ssp5-8.5 and 12% for ssp2-4.5 (Figure S7a and S8a). The model agreement during winter in this region is high (80-95% for both scenarios; Figure S7e and S8e). Consecutive Dry Days will increase in the future in the subtropical Andes with 9 days year$^{-1}$ and 6 days year$^{-1}$ for ssp5-8.5 and ssp2-4.5, respectively (Figure S17c and b). Model projections do not show changes in extreme precipitation events over this area (Figure S19). Precipitation reduction over central Chile is expected to

lead to substantial water deficits and impact hydroelectric outputs and agriculture in the region (Ministry of the Environment,



2015) and our results confirm this. As in the rest of MedClim regions, the increasing risk of water scarcity and drought has triggered National Climate Change Adaptation Plans, which will define public policy for adapting to the long-term effects of warming (Ministry of the Environment, 2015).

*South Africa*

Finally, in South Africa (SAF) winter rainfall fills the large dams around Cape Town that form the core of the Western Cape Water Supply System (Department of Water and Sanitation Republic of South Africa, 2023). Our results point to at least some winter rainfall reduction of 15% for ssp5-8.5 scenario and 12% for ssp2-4.5 scenario (Figure S10a and S11a). The model agreement during winter in this region is high (90-95% for ssp5-8.5 and 80-90% for ssp2-4.5; Figure S10e and S11e). Consecutive Dry Days will increase in the future in Cape Town, 15 days year$^{-1}$ and 9 days year$^{-1}$ for ssp5-8.5 and ssp2-4.5,

respectively (Figure S18c and b). Regarding the maximum one day precipitation, we found that for this area the extreme precipitation event will decrease by 2mm day$^{-1}$ for both scenarios (Figure S20). Our results show the necessity for climate adaptation measures. Cape Town has suffered problems with water management, this was already the case from 2017 to 2018, after three consecutive low rainfall winters, the City of Cape Town faced a severe drought and a water crisis (Parks, 2019). The City of Cape Town municipality successfully managed the water crisis by focusing on reducing water demand, initially

in households and later on in the agriculture sector, with a wide range of measures. The water consumption for the city was reduced from 1,200 million litres in February 2015, to 500 million litres per day in February 2018 (Parks, 2019). One way of communicating the drought was setting a day when the city would "run out of water" named Day Zero. Other water saving measures included restricting available water, new tariffs to penalise excess water usage, water management devices installed in domestic properties, and novel communication strategies (Parks et al, 2019).

To conclude, MedClim regions are familiar with water scarcity due to their climate conditions and increasing water demand that is coupled with economic and population growth. Despite the dry conditions of the regions, they play an important role in global food production due to their climatic conditions. For example, Spain, Italy, Greece, Chile and southern California have strong agricultural sectors that will be at risk, posing economic losses for these countries and affecting food security. According to Food and Agriculture Organization (FAO, 2022), disruptions in production could lead to an increase in food price volatility,

caused by the changing climate. The regions studied have a meridional gradient of precipitation change, the wettest parts located in the north or west of the MedClim regions, provide water to dry areas, usually in the south, through national water plans. However, this is not a long term solution, because our results show that the wet areas of each region will experience a reduction in precipitation in the future under both ssp2-4.5 and ssp5-8.5 scenarios. Other solutions must take into account future precipitation patterns, such as irrigation efficiency (Fader et al., 2016), water reuse and desalination. However,

desalination can be particularly problematic, because it is an energy intensive process and it produces polluting residues (United Nations Environment Programme, 2021), therefore this is only a partial solution. Consequently, measures to reduce net water consumption are needed, in order to alleviate the situation of water scarcity, with policies being most effective in the agriculture and energy sectors (de Roo, 2021).



## 5 Conclusions

Currently, roughly half of the world's population experiences severe water scarcity during at least 1 month per year due to climatic and other conditions (Pörtner, 2022). The climate conditions together with the water demand and geological characteristics determine water availability and water scarcity (European Environmental Agency, 2023). The MedClim regions are already among the regions categorised as having high or extremely high water risk, implying that water demands are increasingly difficult to meet (Water Risk Atlas, 2019).

Climate change is affecting precipitation patterns in all MedClim regions. For both scenarios, ssp2-4.5 and ssp5-8.5, our results show: (1) a decrease in percentage and annual mean precipitation except in CAL; (2) an increase in consecutive dry days; and (3) an increase in maximum one-day and five-day precipitations in CAL, MED and AUS. Even though the study uses CMIP6, the latest generation of models, the uncertainty is still high in the MedClim areas, particularly in CAL, MED and AUS, as already demonstrated by similar studies using CMIP5.

All MedClim regions will experience rainfall changes in the future (2050-2079) due to climate change. Winter precipitation is projected to decrease in all MedClim regions (Figure 3, 4 and Figure S3-12), except for California (Figure 2 and Figure S1 and 2). Even though our results indicate a dominating drying tendency in the MedClim regions, the daily precipitation intensity distribution shows large events for California in all seasons and the Mediterranean Basin during winter, as well as in Southwest of Australia in Autumn (Figure 7). Also, extreme precipitation events that we studied show an increase in all MedClim regions

(Figure 8-10), except in central Chile and South Africa (Figure S17-18). These extreme precipitation events might result in an increased risk of flooding, thus causing serious economic damage. Moreover, all MedClim regions show a tendency of increasing CDD and decreasing CWD in the mid-late century (Figure 11-13 and Figure S19-20). Almost all MedClim regions already experience recurrent droughts, and we show droughts will increase in frequency and intensity, urgeing for mitigation and adaptation measures.

The general reason for the overall increase of dry days in the MedClim regions is the poleward shift in the mid-latitude storm-tracks (Polade et al., 2014). Our results point to a consistent increase of CDD together with the increase of maximum one-day and five-day precipitation, which can have a significant impact on future precipitation in the different MedClim regions. For instance, in MED the majority of the annual precipitation change is determined by the large increase in the number of dry days. Such a tendency will also increase the already substantial interannual precipitation variability in MedClim regions. Therefore,

this climate change driven aridification in the regions will be further exacerbated by the increased year-to-year variability, as a result of the overall less wet days. Water management plans need to be re-considered taking into account the results we provided. Our results demonstrate that it is necessary to implement long-term better than the available solutions to address these challenges, such as integrated water management. It is also important to promote education, awareness and practices about a water saving culture.



### Data availability

Data are publicly available through the websites mentioned in the text:

Copernicus Climate Data Store https://cds.climate.copernicus.eu/#!/home

All the compiled data are available upon contacting the corresponding author (patricia.tarin@isglobal.org).

### Supplement

The supplement related to this article is available

### Author contributions

PTC and IC designed the study. PT performed the main analysis, and together with IC and DP interpreted the results and wrote the manuscript. LCC, JL and XR contributed to the interpretation of the results and to the improvement of the text of the article. All authors approved the final version of the manuscript.

### Competing interests

The authors declare that they have no conflict of interest.

### Acknowledgements

This work was supported by La Caixa Junior Leader Grant 2020 awarded to I. Cvijanovic (Marie Skłodowska-Curie grant agreement No 847648). PTC acknowledges FJC2021-047870-I funded by MCIN/AEI/10.13039/501100011033 and by the European Union NextGenerationEU/PRTR.

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
