# Peer review of "Assessment of Future Precipitation Changes in Mediterranean Climate Regions from CMIP6 ensemble"

_EGUsphere, 2023_

## Community Comment (CC3)

**Comments on:**

**Assessment of Future Precipitation Changes in Mediterranean Climate Regions from CMIP6 ensemble** by Patricia Tarín-Carrasco et al. (2024)

This paper adds to our understanding of the precipitation changes that have occurred in Mediterranean Climate Regions and are likely to occur under anthropogenic climate change. However, the paper lacks some of the chronological history of the research in this area and some of the details for the physical reasons for the observed and future changes. In particular, the paper mentions the work by Polade et al. (2014) but ignores the earlier paper by Frederiksen and Frederiksen (2007), which was the first study to identify the cause of the downwards trend in Southwest Australian rainfall as the change in the Southern Hemisphere (SH) storm tracks. Importantly, they showed that a reduction in the baroclinicity of the SH circulation led to a reduction in the growth rate of the storm tracks modes associated with the rainfall and led to a preference for winter storm development further poleward. Frederiksen et al. (2017) considered changes in the SH storm tracks in all seasons, as well as changes in the baroclinicity and rainfall in both reanalysis and observed datasets, and in a multi-ensemble of CMIP5 models in both the current and future climate scenarios. They also attributed the observed and projected changes to anthropogenic external forcing of the climate.

Two other important papers that are missing from this paper are Frederiksen and Grainger (2015) and Osbrough and Frederiksen (2021). The first of these showed, using observations and CMIP5 models, that the downward trend in Southwest Australian rainfall in the current and future scenarios can be attributed to anthropogenic external forcing. The second paper, using reanalysis data, analysed the interdecadal changes in SH winter explosive storms and South Australian rainfall. They found that decadal changes in the mid-1970s drying of southwestern Western Australia, the Australian Millennium Drought and Southern Australian rainfall anomalies of the early twenty-first century are most related to the intensity of the fast-growing weather systems.

I feel the paper would be improved by the inclusion and discussion of the results contained in these papers.

**References:**

(1) *Interdecadal changes in southern hemisphere winter storm track modes.* Tellus (2007), 59A, 599–617, DOI: 10.1111/j.1600-0870.2007.00264.x

   Jorgen S. Frederiksen and Carsten S. Frederiksen

(2) *The role of external forcing in prolonged trends in Australian rainfall.* Clim Dyn (2015) 45:2455–2468, DOI 10.1007/s00382-015-2482-8

   Carsten S. Frederiksen and Simon Grainger

(3) *Trends and projections of Southern Hemisphere baroclinicity: the role of external forcing and impact on Australian rainfall.* Clim Dyn (2017) 48:3261–3282, DOI 10.1007/s00382-016-3263-8

   Carsten S. Frederiksen, Jorgen S. Frederiksen, Janice M. Sisson, Stacey L. Osbrough

(4) *Interdecadal changes in Southern Hemisphere winter explosive storms and Southern Australian rainfall.* Clim Dyn (2021) 56:3103–3130, DOI: 10.1007/s00382-021-05633-y

Stacey L. Osbrough and Jorgen S. Frederiksen